# Prioritizing interventions for cholera control in Kenya, 2015–2020

**Waqo Boru**[1,2☯], **Shaoming Xiao**[3☯], **Patrick Amoth**[1], **David Kareko**[1], **Daniel Langat**[1], **Ian Were**[1], **Mohammad Ali**[5], **David A. Sack**[5], **Elizabeth C. Lee**[4‡], **Amanda K. Debes**[5‡]*

**1** Ministry of Health, Nairobi, Kenya, **2** Field Epidemiology and Laboratory Training Program, Nairobi, Kenya, **3** Johns Hopkins University School of Medicine, Baltimore, Maryland, United States of America, **4** Johns Hopkins School of Public Health, Department of Epidemiology, Baltimore, Maryland, United States of America, **5** Johns Hopkins School of Public Health, Department of International Health, Baltimore, Maryland, United States of America

☯ These authors contributed equally to this work.
‡ ECL and AKD also contributed equally to this work.
* adebes1@jhu.edu

**Data Availability Statement:** We have submitted the cleaned data used in this analysis as S1 Table.

**Funding:** Research reported in this publication was supported in part by the Bill & Melinda Gates Foundation (BMGF) under award numbers

## Abstract

Kenya has experienced cholera outbreaks since 1971, with the most recent wave beginning in late 2014. Between 2015–2020, 32 of 47 counties reported 30,431 suspected cholera cases. The Global Task Force for Cholera Control (GTFCC) developed a *Global Roadmap for Ending Cholera by 2030*, which emphasizes the need to target multi-sectoral interventions in priority cholera burden hotspots. This study utilizes the GTFCC's hotspot method to identify hotspots in Kenya at the county and sub-county administrative levels from 2015 through 2020. 32 of 47 (68.1%) counties reported cholera cases during this time while only 149 of 301 (49.5%) sub-counties reported cholera cases. The analysis identifies hotspots based on the mean annual incidence (MAI) over the past five-year period and cholera's persistence in the area. Applying a MAI threshold of 90th percentile and the median persistence at both the county and sub-county levels, we identified 13 high risk sub-counties from 8 counties, including the 3 high risk counties of Garissa, Tana River and Wajir. This demonstrates that several sub-counties are high level hotspots while their counties are not. In addition, when cases reported by county versus sub-county hotspot risk are compared, 1.4 million people overlapped in the areas identified as both high-risk county and high-risk sub-county. However, assuming that finer scale data is more accurate, 1.6 million high risk sub-county people would have been misclassified as medium risk with a county-level analysis. Furthermore, an additional 1.6 million people would have been classified as living in high-risk in a county-level analysis when at the sub-county level, they were medium, low or no-risk sub-counties. This results in 3.2 million people being misclassified when county level analysis is utilized rather than a more-focused sub-county level analysis. This analysis highlights the need for more localized risk analyses to target cholera intervention and prevention efforts towards the populations most vulnerable.

OPP1148763 (DS) and INV-002667 (EL) and in part through a grant from the National institute of Allergy and Infectious Disease (NIAID) under award number 5R01AI123422 (DS). The funders had no role in study design, data collection and analysis, decision to publish, or preparation of the manuscript.

**Competing interests:** The authors have declared that no competing interests exist.

## Author summary

Kenya has experienced recurrent cholera outbreaks from 1971 through today and constraints on resources make it essential to target multi-sectoral interventions to high priority areas. Using surveillance data from 2015–2020, this study identifies high priority areas for cholera intervention in Kenya following guidance from the Global Task Force on Cholera Control. We identified that roughly 3 million people (6% of the population) in 13 sub-counties live in high priority areas in Kenya, where the mean annual incidence exceeded 70 cases per 100,000 population and over 35% of weeks reported at least one suspected cholera case. Further, 1.6 million people living in high priority sub-counties would have been de-prioritized had the analysis been performed only at the county scale. Cholera interventions in Kenya should target high priority sub-counties, and future countries undergoing cholera control planning should carefully consider operational implementation when determining the spatial scale of their prioritization analysis.

## Introduction

Cholera remains a public health threat to over one billion people globally [1–3], and the majority of cases reported to the World Health Organization (WHO) are from sub-Saharan Africa. [4] Cholera was first reported in Kenya in 1971 and Kenya has since experienced outbreaks regularly every few years, with the most recent wave of outbreaks beginning in December 2014 [5–7]. Kenya reported the "index case" of the epidemic wave onset in Eastern and Southern Africa in December 2014, which spread to other African nations including Tanzania, Uganda and South Sudan [8]. As such, cholera control efforts in Kenya not only protect vulnerable people in Kenya, but could also stop spread to neighboring countries.

To combat cholera in Kenya, the Ministry of Health (MoH) developed a national cholera control plan based on the guidance from the WHO-led Global Task Force on Cholera Control (GTFCC). The GTFCC's *Global Roadmap for Ending Cholera by 2030* [9] stresses the importance of targeting control activities to priority intervention areas, also known as "cholera burden hotspots." These are administrative units with high incidence rates, persistent cholera transmission, or high risk of cholera introduction relative to other locations in the country [9–11]. Identification of these high priority areas can help with efficient targeting of oral cholera vaccination (OCV) campaigns, surveillance system investments, and water and sanitation infrastructure, all of which could be used to coordinate a comprehensive cholera control strategy within the country.

In this analysis, we identified priority intervention areas for cholera using data from the Kenya MoH from 2015–2020. We performed this analysis at the county and sub-county administrative units following the guidance issued by GTFCC.[12]

## Methods

### Cholera surveillance data

Notifiable diseases are reported via national line list data from county and sub-county health facilities to the Kenya MoH as part of the Integrated Disease Surveillance and Response (IDSR) program [13]. We examined all suspected cholera cases across Kenya from the national line list data from January 1, 2015 through Dec 31, 2020. The study team cleaned location names and performed name disambiguation to match sub-county names to Global Administrative Areas (GADM) database of worldwide shapefiles sub-county locations [14]. For sub-

county names with no corresponding label on the map, the names were cross-referenced with sub-counties listed in the *Kenya 2019 Population and Housing Census Report Volume 2* (2019 Census) [15] and subsequently aligned with the map. In Kenya, there are 47 counties, with an average population size of 1.1 million in 2019, and 301 sub-counties, with an average population size of 176,000 in 2019. Further data cleaning details are described in S1 Fig.

IDSR guidelines define a suspected cholera case as the sudden onset of three or more episodes of watery diarrhea within 24 hours in a person $\geq 2$ years of age or a younger patient in whom a clinician suspected cholera. A confirmed cholera case was defined as a suspected case that was confirmed by culture or PCR with *Vibrio cholerae* O1 or O139. Cases were assigned to the date they sought care at a health facility, or if missing, the date of diarrhea onset.

## Population data

The annual population counts from 2015–2020 were aggregated from 100-meter-resolution population raster of Kenya to sub-county and county administrative units [16,17] using the R package "exactextractr" (version 0.6.1) [18]. Sub-county and county-level administrative maps of Kenya were obtained from the GADM database of worldwide shapefiles [14].

## Prioritizing cholera burden hotspots

The prioritization exercise was performed separately at the county and sub-county units of analysis. For each geographic scale, we identified cholera burden hotspots, also known as priority intervention areas, according to the GTFCC-recommended method [12], which uses two quantitative indicators: 1) mean annual incidence (MAI), the mean of the annual cholera incidence rate (cases per 100,000 population) over the analysis period; 2) persistence, the percentage of weeks in the analysis period where at least one cholera case was reported [12],

We identified thresholds for each indicator, and locations were categorized as high priority if both indicators exceeded their thresholds and medium priority if only one indicator exceeded its threshold. Locations that remained below both MAI and persistence thresholds were low priority. Based on discussions with the MoH regarding the feasibility, cost, and logistics for targeting interventions across the country and the relative importance of each metric to country priorities, we selected the 90[th] percentile value as the MAI threshold (33 per 100,000) and the median value as the persistence threshold (2.88%) at the sub-county scale. We also performed sensitivity analyses for all combinations of MAI and persistence thresholds in decile increments; a subset of these is displayed in the Supplementary Material.

## Software used

The software programs used include Microsoft Excel for data management, ArcGIS 10.6 for the management of geographic information files and mapping, and R (version 4.1.0) for statistical computing and graphics.

## Ethics

The study used de-identified public health disease surveillance data. Therefore, no ethical approval was required for conducting this study.

## Results

From 2015 to 2020, the total annual suspected cases ranged from a maximum of 9464 in 2015 to a minimum of 711 in 2020, with 32 of 47 counties reporting 30,431 cases overall. After excluding records with missing date or missing or unidentifiable geographic information, we

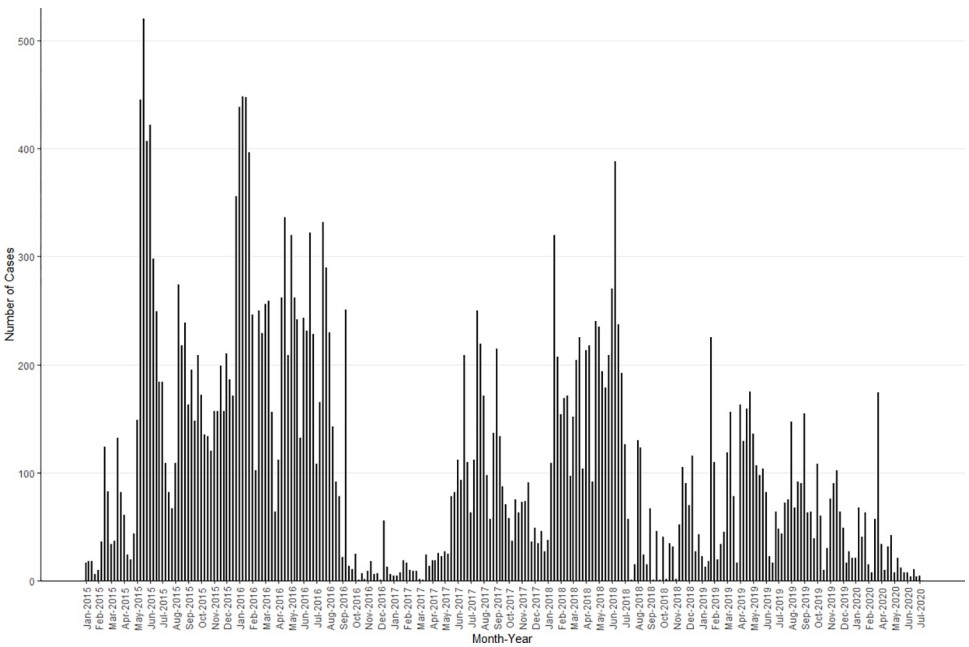

**Fig 1. Weekly suspected cholera cases in Kenya from 2015–2020.** No more cases were reported in 2020 after July so the figure is truncated after that point.

successfully geolocated 29,082 cases to counties and 22,825 to sub-counties, and these were the final datasets used for the county and sub-county analyses, respectively (See additional information on data cleaning in Fig A in S1 Fig).

Cholera cases varied by month and year, and while there was not a readily discernable seasonal or cyclical pattern, there were extended periods with relatively high cholera transmission (May 2015-August 2016, June 2017-July 2018, and February 2019-November 2019) interspersed with months of relatively sparse or low activity (Fig 1). Out of the 32 counties affected during this 6-year period, 1 county (Garissa) reported cases in all 6 years, 3 counties reported cases in 5 of 6 years, 5 counties reported cases in 4 of the 6 years, 6 counties reported cases in 3 of the 6 years, 11 counties reported cases in 2 of the 6 years, and 6 reported cases in only one year. The sub-counties with the highest case count and incidence varied by year, suggesting that outbreaks in different regions of the country contributed to the periods with relatively high transmission (Figs B2 in S1 Text and 2).

Mean annual incidence (MAI) and persistence were calculated for each sub-county in order to perform the prioritization activity according to the 90th percentile and median thresholds, respectively (Figs 3–5 and C in S1 Text). The sub-counties with high mean annual incidence tended to have high persistence, and few locations exceeded the 90th percentile MAI while remaining below the median persistence value. Thirteen sub-counties (6% of the population) were categorized as high priority, 61 sub-counties (24% of the population) as medium priority, and 76 sub-counties (25% of the population) as low priority (Tables 1 and Table E in S1 Text). There were 151 sub-counties (45% of the population) that had no reported cases in the analysis period.

When performing the same prioritization analysis at the county scale instead of the sub-county scale, classification becomes much less efficient due to reduced resolution (Fig 6). While the county-level analysis identified three high priority counties in Northeast Kenya (Garissa, Tana River and Wajir), the sub-county analysis identified thirteen high priority sub-

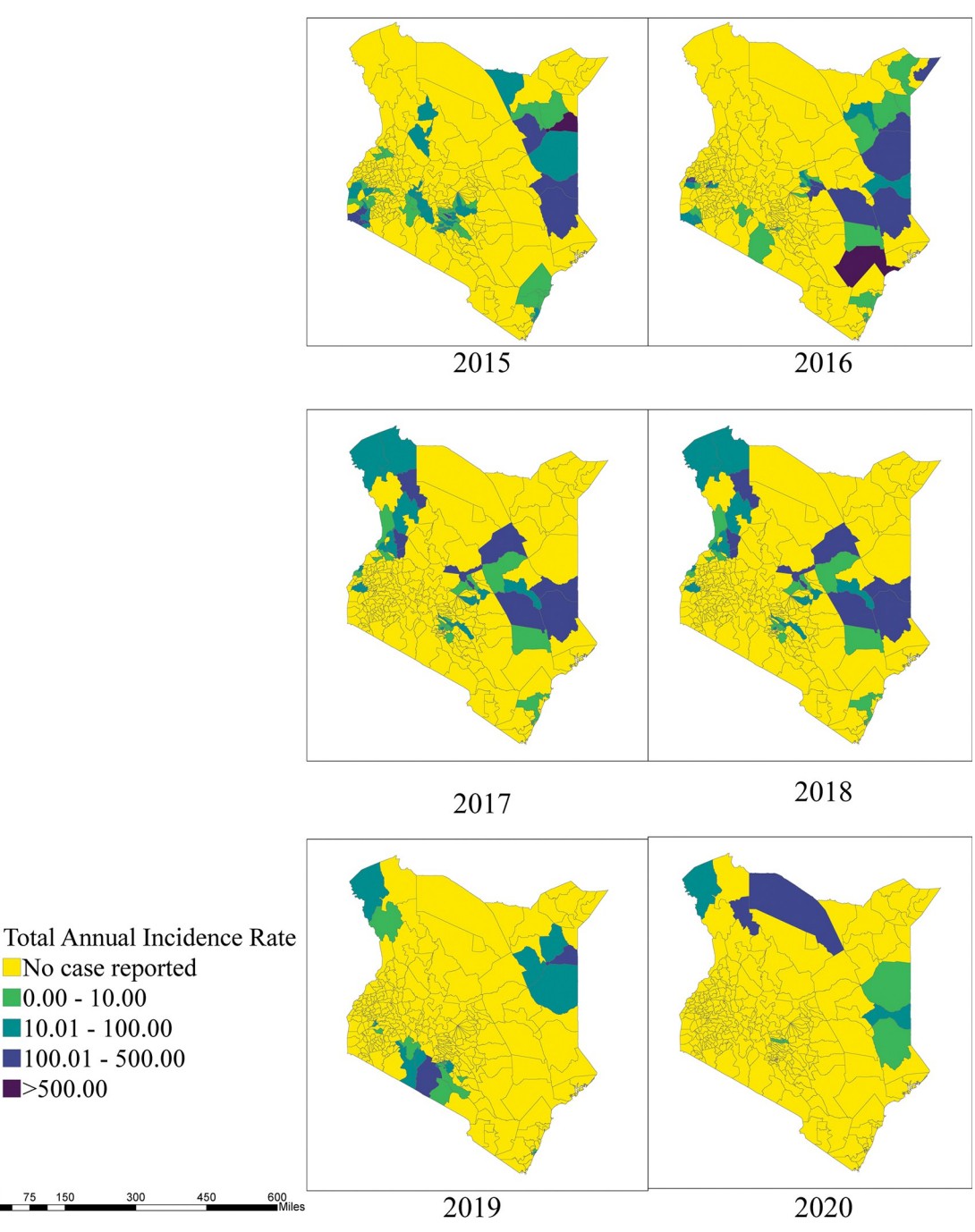

**Fig 2. Total annual cholera incidence rate by sub-counties, Kenya, 2015–2020.**

counties from eight counties, including those in Nairobi County and five additional counties that border neighboring countries. When finer resolution data are not available, significant misclassification in both directions may occur (Table 2). For example, roughly 660,000 individuals living in sub-counties with no reported cases were classified as living in a high priority county, and 1.6 million living in high priority sub-counties were classified as living in a medium priority county. While no populations living in high priority sub-counties would be

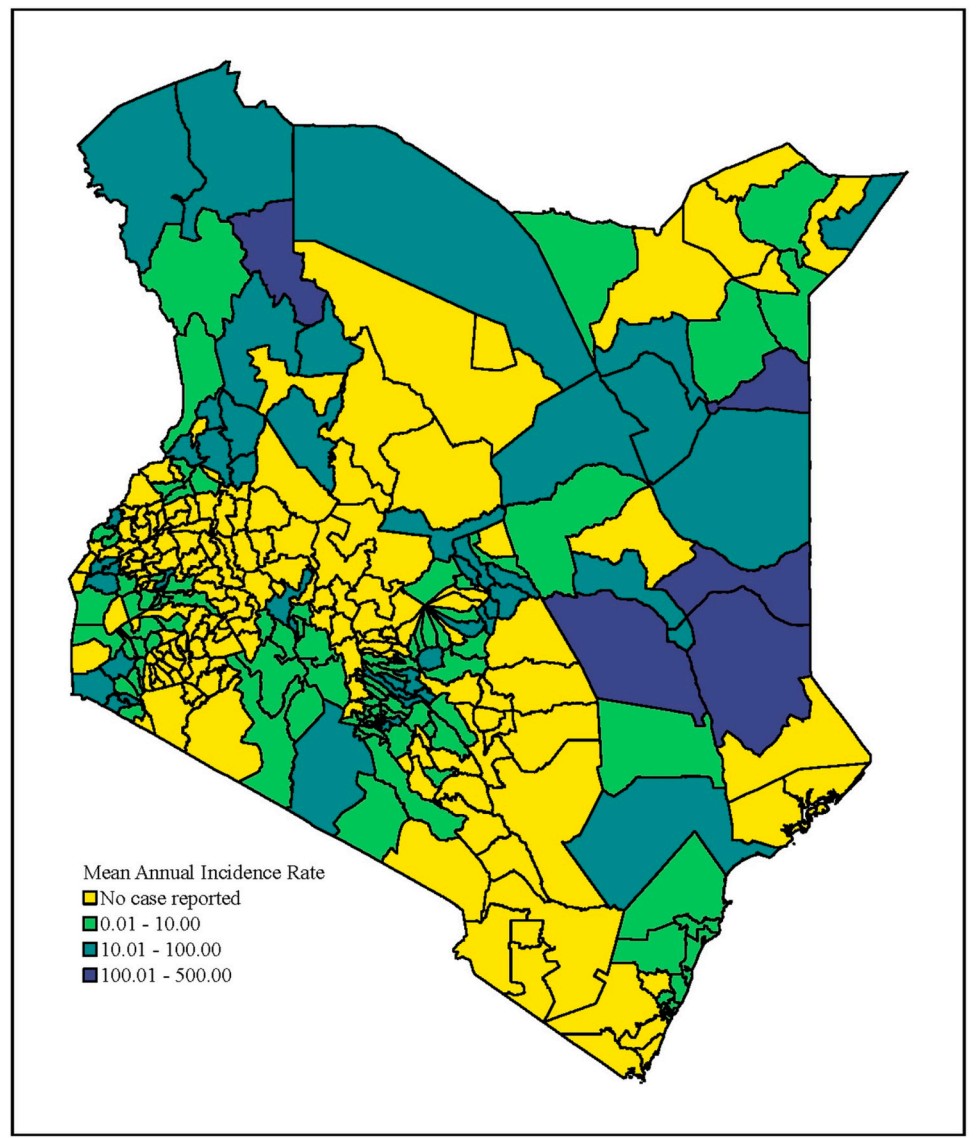

**Fig 3. Mean annual incidence in Kenya by sub-county from 2015–2020.**

classified as living in low priority counties, there were would be a significant "over-prioritizations" of populations living in sub-counties with no or few cases reported.

In the county-level analysis, only 4,025 out of 29,082 suspected cases (14%) were tested for culture, among which 2,525 (63%) were positive; only 3,327 RDTs were performed (11%), among which 82% were RDT-positive. Among the 3 high priority counties, Tana River reported 30% of RDT positivity among those tested (36/120), while Wajir reported a 93% positivity rate among those tested (591/633). All 3 counties reported similar rates of culture positive when comparing across all counties, with Garissa (591/867) and Tana River (11/16) reporting 68% culture positive among those tested and Wajir reporting 74% culture positive (345/461) (Table F in S1 Text).

Due to challenges with geolocation, we assessed the sensitivity of the results when making different location name cleaning assumptions. There were many records that could only be

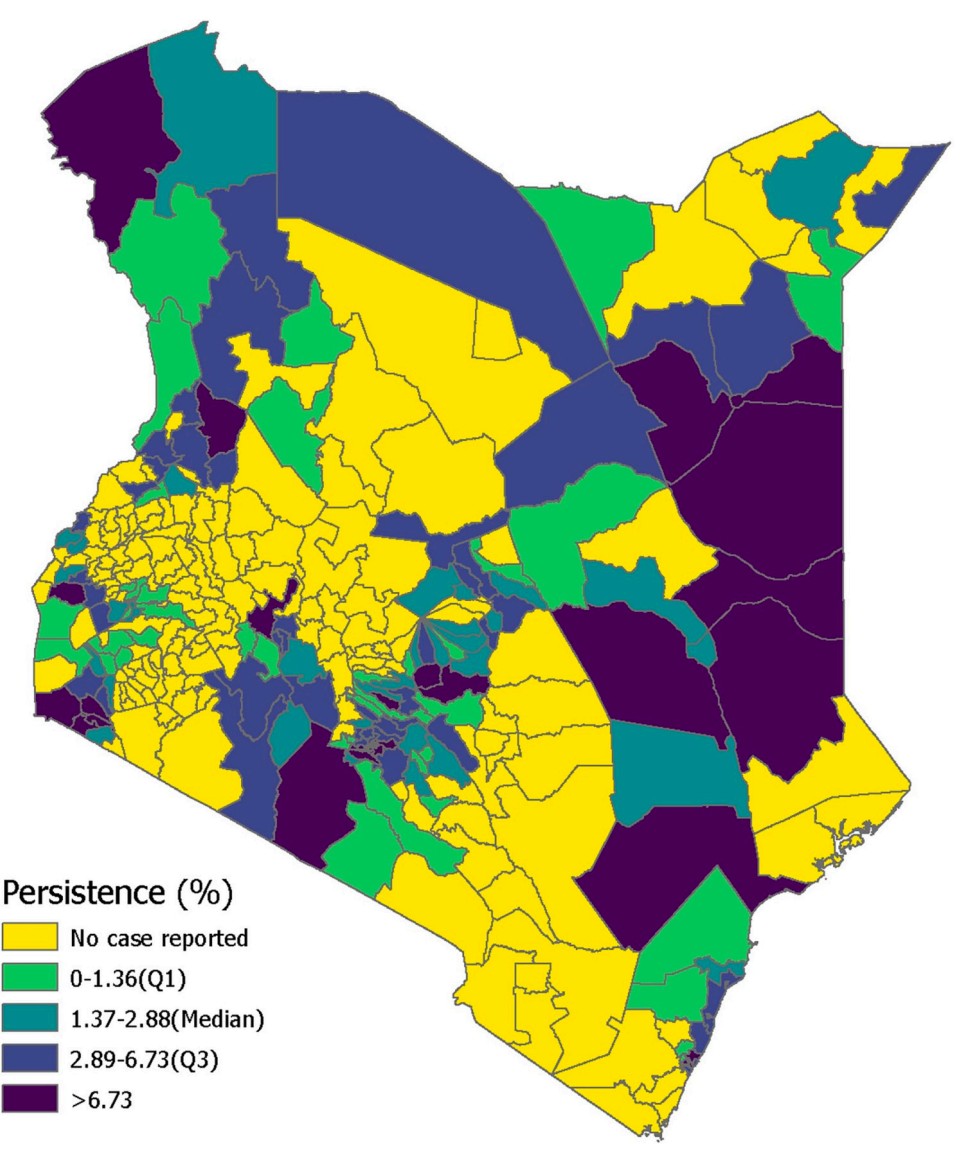

**Fig 4. Persistence in Kenya by sub-county from 2015–2020.**

geolocated to the county scale, as the sub-county field was missing or otherwise unidentifiable. To assess the sensitivity of our hotspots to these missing data, we performed a separate analysis on a sub-county dataset, where in one arm we excluded records that had insufficient data to geolocate the case to a specific sub-county. In the comparison arm, we included those same non-geolocatable records by assigning them proportionally according to the distribution of geolocatable sub-county records in the associated county and reporting year. The sensitivity analysis showed the same high-risk hotspots in both arms of the analysis, identifying over 3 million people at high-risk regardless of which data set is used for the analysis (Fig D and Table C in S1 Text).

We also explored the impact of alternate thresholds for MAI (Fig E and Table E in S1 Text). For example, when reducing the primary analysis's 90th percentile threshold for MAI to the 80th percentile threshold, 11% or over 5 million people lived in priority areas (compared to the 6% or roughly 3 million people in the primary sub-county analysis).

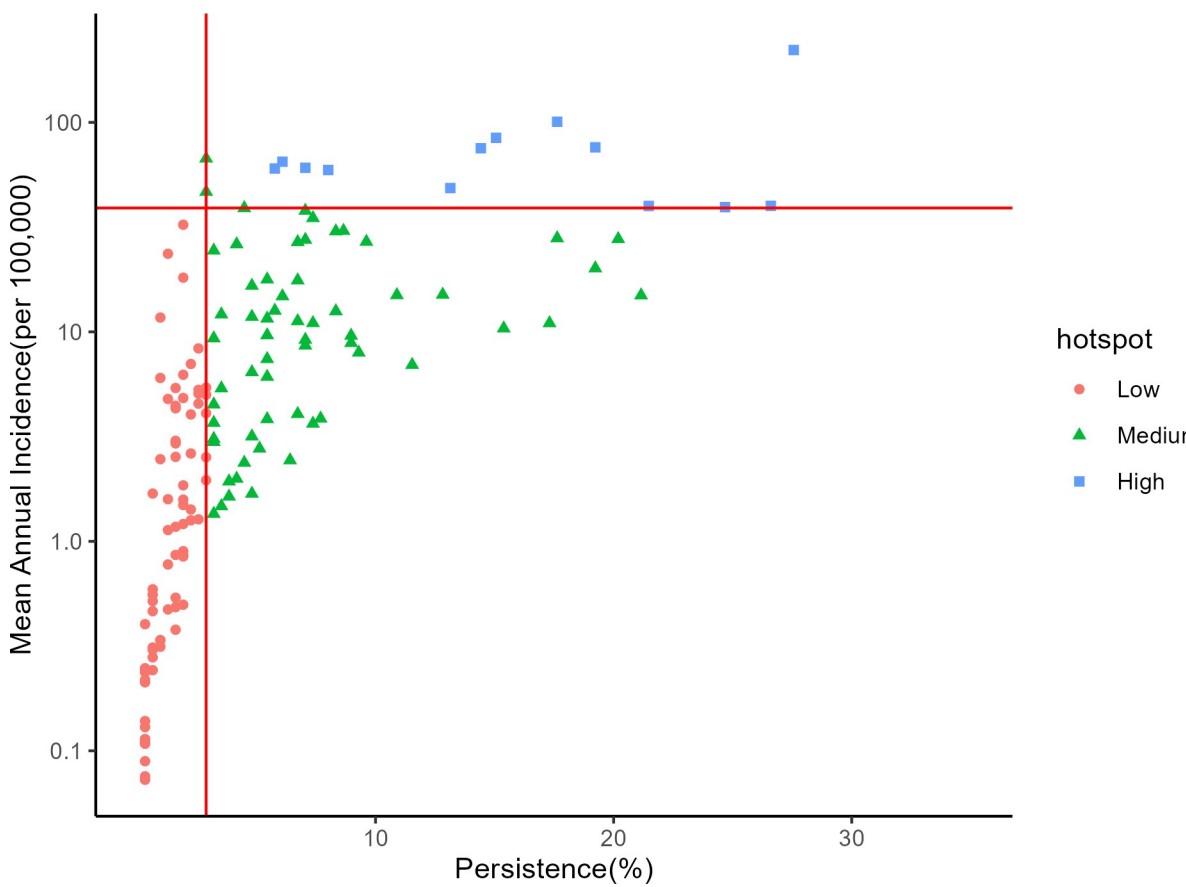

**Fig 5. Scatterplot for mean annual incidence (on the log scale) versus persistence, the percentage of total weeks reporting at least one suspected cholera case.** The horizontal and vertical lines represent the mean annual incidence and persistence thresholds, respectively.

## Discussion

In this analysis we collated cholera IDSR line list data from 2015 to 2020 in Kenya to identify priority intervention areas at both county and sub-county scales following the proposed GTFCC method. At the sub-county scale, there were 13 high priority areas from a total of 8 counties that represented 6% of the Kenyan population. When comparing the same analyses at the county and sub-county scales, we found that sub-counties were more likely to be over-prioritized (given a higher priority at the county scale than what they would have received at the sub-county scale) than under-prioritized when data were analyzed in the same way at the county scale. However, high priority sub-counties were relatively stable under different cleaning methods and threshold settings.

**Table 1. Population and number of sub-counties under each hotspot (priority) level.**

| Hotspot (Priority) Level | Population | Percentage of population | Number of sub-counties | Population Weighted Average MAI (cases per 100,000): Mean (Min, Max) | Population Weighted Average Persistence Mean (Min, Max) |
|---|---|---|---|---|---|
| High | 3021238 | 6 | 13 | 74.02 (39.39, 221.55) | 36.60 (25.00, 47.12) |
| Medium | 12124262 | 24 | 61 | 12.25 (1.35, 67.26) | 20.78 (7.69, 41.67) |
| Low | 12559717 | 25 | 76 | 2.35 (0.07, 32.46) | 5.06 (0.64, 9.29) |
| No case reported | 23041691 | 45 | 151 | 0 | 0 |

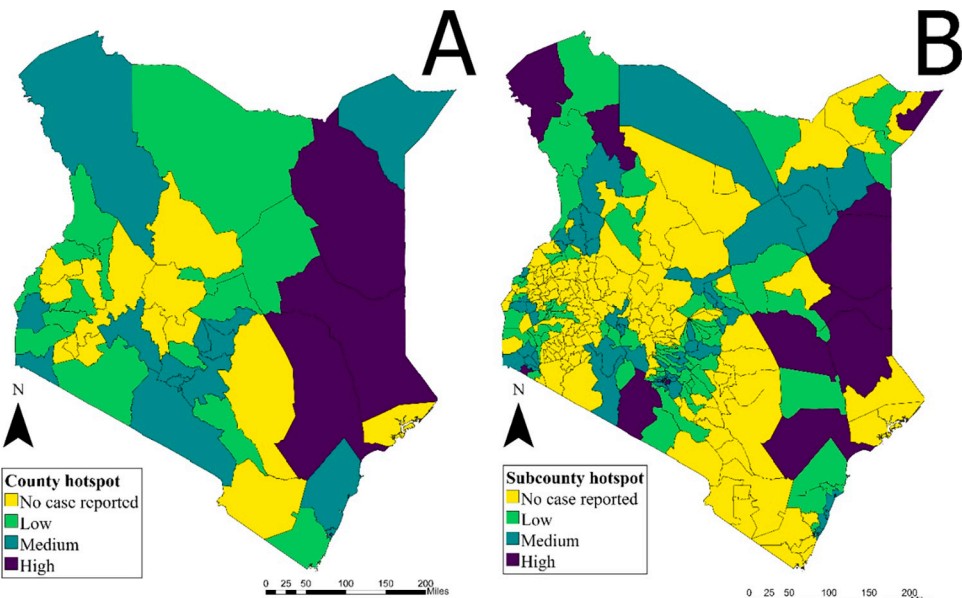

**Fig 6.** A) Priority intervention areas at the county scale in Kenya 2015–2020 (MAI threshold: 90 percentile– 33 per 100,000, Persistence threshold: median– 9.78%). B) Priority intervention areas at the sub-county scale in Kenya from 2015–2020 (MAI threshold: 90 percentile– 39 per 100,000, Persistence threshold: median– 2.88%).

Previous work has reported that Kenya has early-year (e.g. January) and relatively low amplitude cholera seasonality, and that there were several cholera outbreaks in the country during the 2015–2019 period [19]. The four seasons in Kenya include a dry season from January to March, a rainy season from March to May, a dry season from May to October and a rainy season from October to December [20]. According to the 2009 census, and more recently the 2021 Progress on Household Drinking Water, Sanitation and Hygiene produced by the JMP, ~90% of urban households have access to improved water sources, while only approximately 50% of rural households have such access [20,21] In addition, the type of improved water source greatly differs, with urban households predominantly having piped water into the dwelling while rural households primarily have access to dug wells as an improved water source. Access to adequate sanitation is also a significant issue in Kenya, with the 2009 census reporting that only 30% of urban and 20% of rural households have access to an improved toilet facility [20]. The rural population of Kenya comprises 68.8%, or 32,732,596 people, out of the total population of just over 47million people [22]. Systematic collection of risk factor data by sub-county would improve understanding of the role of risk factors in high-risk areas and facilitate intervention decisions.

**Table 2. Comparison of population living in different priority areas in the county and sub-county analyses.** When data are not available at the finer resolution scale, classification of priority areas becomes less efficient. For example, roughly 660,000 people living in sub-counties with no reported cases would be categorized as living in high priority counties.

| | | County Hotspot | | | |
| --- | --- | --- | --- | --- | --- |
| | | No case reported | Low | Medium | High |
| Sub-county Hotspot | No case reported | 13859714 | 5768177 | 2755295 | 658505 |
| | Low | 0 | 5112447 | 7015429 | 431841 |
| | Medium | 0 | 2638988 | 8933536 | 551738 |
| | High | 0 | 0 | 1626394 | 1394844 |

The current GTFCC method for priority intervention areas recommends that cholera data are collated at an administrative level 2 (e.g. districts) or smaller level to facilitate operational planning. At the request of the MoH, this analysis was performed at the sub-county level (admin level 3 in Kenya as county is admin level 2). The counties in Kenya are comprised of populations with a median size greater than 1 million people, thus, it would be challenging to target interventions at the county scale. By focusing on the sub-county, targeting becomes more efficient, and the population sizes are reduced from populations in the millions to those in the low hundreds of thousands, increasing the feasibility and likely impact of interventions. Further, when Kenya introduces OCV, county level vaccination targets are likely both too large for the vaccine stockpile to consider and too challenging logistically. A recent hotspot analysis in Uganda identified districts as hotspots, but based on local knowledge of the county officials, specific sub-counties were identified as the areas at highest risk and these sub-counties were then targeted for OCV [23]. Similarly, an analysis of hotspots in Nigeria showed that within the LGAs (Local Government Area) (analogous to a district), certain wards were at higher risk than other wards [24]. These two studies highlight the importance of considering the spatial scales of surveillance and implementation, as they may impact the efficiency, cost-effectiveness, and public health impact of disease control measures.

There were several limitations to this study. While the IDSR program reports cholera line lists at the county level, sub-county level reporting is not as consistent. We used sub-county level data reported to the MoH for this analysis, however only 22,825 records out of the total 30,431 line list records were able to be cleaned and their sub-counties identified. Further, the county and sub-county datasets were different due to location name data cleaning challenges, which means that these analyses are not exactly comparable. Our analysis may have dropped records from the same, repeated location names that could not be identified successfully, which would bias the spatial distribution of cases and the subsequent results. Another limitation is due to reporting of cases on the basis of clinical presentation. The outbreaks were initially confirmed as cholera by culture, but subsequent cases were reported based on clinical criteria. Some of the cases reported as cholera may have had illness due to another pathogen, and conversely, cholera cases may be under-reported, especially in remote areas. As the use of low-cost and effective rapid diagnostic tests becomes more pervasive in cholera surveillance, it would be valuable to consider cholera test positivity as an additional indicator when prioritizing areas for intervention [25].

This analysis may provide information useful to the Kenya MoH to support their National Cholera Control Plan and intervention efforts, including the use of OCV, in targeting cholera control in the high-risk sub-counties identified through this analysis. The cholera hotspot mapping results were shared with the MOH Kenya to support the preparation of the national cholera control plan. It is the hope that the hotspot results provide an evidence-based approach to prioritize high risk sub-counties in the implementation of the available interventions. Further, the results emphasize the need to conduct situation specific analyses to understand contextual factors driving cholera outbreaks to inform interventions as well as highlights the need to digitize cholera surveillance tools, including case reporting and monitoring of WASH indicators. In addition, we hope that documentation of the methods of this exercise could aid other countries engaged in national cholera control planning.

## Supporting information

**S1 Text.** Fig A in S1 Text: Data Cleaning Flow Chart. Fig B in S1 Text: Total annual cholera cases by sub-counties, Kenya, 2015–2020. Fig C in S1 Text: Scatterplot for Mean Annual Incidence versus persistence (Sensitivity Analysis). Fig D in S1 Text. Priority intervention areas at

the sub-county scale, using a different location name cleaning methodology–Sensitivity Analysis (MAI threshold: 90 percentile– 31.4 per 100,000, Persistence threshold: median– 5.45%). Fig E in S1 Text: Sensitivity analysis for priority intervention areas at the sub-county scale after assuming different thresholds for mean annual incidence. The persistence threshold remained constant at the median value of 2.88%. Table A in S1 Text: Number of missing subcounty observations by county and year. Table B in S1 Text. Population and number of sub-counties under each hotspot level (sensitivity analysis) based on MAI threshold: 90 percentile– 39 per 100,000, Persistence threshold: median– 2.88%. Table C in S1 Text. Hotspot sensitivity transition matrix. Presents sub-county hotspot classification change based on data cleaning. The Clean Data Hotspot are records that had insufficient data to categorize the case, and as a result, were exluded from the Analysis. The Sensitivity Analysis Hotspot takes the same record with insufficient data and uses extrapolation to assign the sub-county and county information. Table D in S1 Text. Table provides % population residing in a defined hotspot area based on MAI threshold. Table E in S1 Text. Subcounty by Priority. Table F in S1 Text. Test Summary by County by Year.
(DOCX)

**S1 Table. Hotspot Analysis Data.**
(XLSX)

## Author Contributions

**Conceptualization:** Waqo Boru, Shaoming Xiao, David Kareko, Ian Were, Mohammad Ali, David A. Sack, Elizabeth C. Lee, Amanda K. Debes.

**Data curation:** Waqo Boru, Shaoming Xiao, David Kareko, Daniel Langat, Ian Were, Mohammad Ali, Elizabeth C. Lee, Amanda K. Debes.

**Formal analysis:** Waqo Boru, Shaoming Xiao, Mohammad Ali, David A. Sack, Elizabeth C. Lee, Amanda K. Debes.

**Funding acquisition:** David A. Sack, Elizabeth C. Lee, Amanda K. Debes.

**Investigation:** Waqo Boru, Shaoming Xiao, Daniel Langat, David A. Sack, Elizabeth C. Lee, Amanda K. Debes.

**Methodology:** Waqo Boru, Shaoming Xiao, David Kareko, Mohammad Ali, David A. Sack, Elizabeth C. Lee, Amanda K. Debes.

**Project administration:** Patrick Amoth, Daniel Langat, Ian Were, David A. Sack, Elizabeth C. Lee, Amanda K. Debes.

**Resources:** Waqo Boru, Patrick Amoth, Daniel Langat, Ian Were, Elizabeth C. Lee.

**Supervision:** Patrick Amoth, Daniel Langat, Ian Were, David A. Sack, Elizabeth C. Lee, Amanda K. Debes.

**Validation:** Elizabeth C. Lee, Amanda K. Debes.

**Visualization:** David A. Sack, Elizabeth C. Lee.

**Writing – original draft:** Waqo Boru, Shaoming Xiao, Patrick Amoth, David Kareko, Daniel Langat, Ian Were, Mohammad Ali, David A. Sack, Elizabeth C. Lee, Amanda K. Debes.

**Writing – review & editing:** Waqo Boru, Shaoming Xiao, Patrick Amoth, David Kareko, Daniel Langat, Ian Were, Mohammad Ali, David A. Sack, Elizabeth C. Lee, Amanda K. Debes.

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
