## [Decision Letter · Decision Letter 0]

25 Jan 2023

Dear Dr. Debes,

Thank you very much for submitting your manuscript "Prioritizing interventions for cholera control in Kenya, 2015-2020" for consideration at PLOS Neglected Tropical Diseases. As with all papers reviewed by the journal, your manuscript was reviewed by members of the editorial board and by several independent reviewers. The reviewers appreciated the attention to an important topic. Based on the reviews, we are likely to accept this manuscript for publication, providing that you modify the manuscript according to the review recommendations. 

Sincerely,

Elsio Wunder Jr, D.V.M., Ph.D.

Section Editor

Elsio Wunder Jr

Section Editor

Reviewer's Responses to Questions

**Key Review Criteria Required for Acceptance?**

**Methods**

-Are the objectives of the study clearly articulated with a clear testable hypothesis stated?

-Is the study design appropriate to address the stated objectives?

-Is the population clearly described and appropriate for the hypothesis being tested?

-Is the sample size sufficient to ensure adequate power to address the hypothesis being tested?

-Were correct statistical analysis used to support conclusions?

-Are there concerns about ethical or regulatory requirements being met?

Reviewer #1: The reviewer found the objective of the study to be clear. There were a few issues that would improve the readers understanding of the methods. (1) The IDSR definition of suspect cholera (3 or more episodes in 24 hours, age>2) appears to have low specificity. ETEC, Campy, Shigella infections would also cause 3 or more episodes in 24 hours for children and adults. Can the authors provide data supporting the validity of this definition for case detection? Was there a requirement that suspects cases be epidemiologically linked to a laboratory confirmed case? (2) Provide additional information on how the 90th percentile for MAI was chosen. Did you rank all MAI by county and select the 90th percentile value? Provide methods for selecting the persistence cutoff too. (3) For readers unfamiliar with Kenya, it would be helpful to describe the administrative division of the country. What is the number of counties and subcounties and number rural and urban counties/subcounties. How do counties/subcounties differ by population size. It may be helpful to give an example of a county with total number of subcounties. (4) In figures, it was unclear what the denominator was for "Total Annual Incidence Rate." Was this the number of cases or case number per population value (e.g., 100,000 persons)? (5) The authors identified both suspect and confirmed cholera cases. Can the authors provide data on the frequency of laboratory confirmed cases in high incidence areas?

Reviewer #2: (No Response)

Reviewer #3: Methods

All clear. 

Line 99: Could you specify who discussions at the MOH were held with e.g. departments? How many people? And what you mean by “logistics for targeting interventions”? I think it is a very important point that targeting is based on the ability and resources available within the country and this may affect the thresholds set by other countries. For instance, Kenya have said 90th percentile but other settings may need to specify higher or lower based on their capacities. If you could go into 1-2 sentences more detail on the decision making, this would be useful.

**Results**

-Does the analysis presented match the analysis plan?

-Are the results clearly and completely presented?

-Are the figures (Tables, Images) of sufficient quality for clarity?

Reviewer #1: This reviewer found the analysis and presentation of results to be clear.

Reviewer #2: (No Response)

Reviewer #3: Results

All clear. 

Table 1: is there a full list of the sub-counties included in the high, medium and low priority levels? I would find this useful as a planning tool.

**Conclusions**

-Are the conclusions supported by the data presented?

-Are the limitations of analysis clearly described?

-Do the authors discuss how these data can be helpful to advance our understanding of the topic under study?

-Is public health relevance addressed?

Reviewer #1: This is a very practical study with implications for cholera control in and outside Kenya. The limitations section was adequate. In the limitations section, the authors addressed my concern with the cholera case definition. It would be appropriate to further explore in brief what are the implications for low specificity. The authors should consider a brief discussion on alternative, if any, to this approach to selecting target areas. Are there other approaches that an MoH could consider.

Reviewer #2: (No Response)

Reviewer #3: Although the authors have talked about how counties and sub-counties would be prioritised during 2015-2020, based on the data available, it would be useful to see what the authors consider recommendations going forward. Should these 13 priority areas from 8 counties be prioritised from 2020 onwards? Could the authors add more to the discussion on what the implications are for control measures? Did the authors re-consult the MoH on what this means for Kenya’s cholera strategy?

**Editorial and Data Presentation Modifications?**

Reviewer #1: GADM database was mentioned. In a sentence, clarify what is the GADM. Many readers will not know.

Reviewer #2: (No Response)

Reviewer #3: n/a

**Summary and General Comments**

Reviewer #1: This is a very practical study with important implications for cholera control in cholera endemic countries. The reviewer mentioned several methodological issues that should be addressed. Beyond these issues, the author does not have any concerns.

Reviewer #2: (No Response)

Reviewer #3: Overall: 

A very interesting paper and analysis of existing data! This paper is a good example of how countries can use their own surveillance systems to prioritise and target cholera control resources. 

It would be useful to see recommendations based on this paper included in the discussion. This paper is led by the MoH and it could be an opportunity for them to suggest a plan based on the findings of this paper. Other minor comments below.

PLOS authors have the option to publish the peer review history of their article (what does this mean?). If published, this will include your full peer review and any attached files.

Reviewer #1: No

Reviewer #2: Yes: Aminata P Kilungo

Reviewer #3: Yes: Lauren D'Mello-Guyett

Figure Files:

Data Requirements:

Reproducibility:

References

---

## [Editor Report · Decision Letter 1]

23 Mar 2023

Dear Dr. Debes,

Thank you very much for submitting your manuscript "Prioritizing interventions for cholera control in Kenya, 2015-2020" for consideration at PLOS Neglected Tropical Diseases. As with all papers reviewed by the journal, your manuscript was reviewed by members of the editorial board and by several independent reviewers. The reviewers appreciated the attention to an important topic. Based on the reviews, we are likely to accept this manuscript for publication, providing that you modify the manuscript according to the review recommendations. 

Sincerely,

Elsio Wunder Jr, D.V.M., Ph.D.

Section Editor

Figure Files:

Data Requirements:

Reproducibility:

References

---

## [Editor Report · Decision Letter 2]

20 Apr 2023

Dear Dr. Debes,

We are pleased to inform you that your manuscript 'Prioritizing interventions for cholera control in Kenya, 2015-2020' has been provisionally accepted for publication in PLOS Neglected Tropical Diseases.

Best regards,

Elsio Wunder Jr

Section Editor

---

## [Editor Report · Acceptance letter]

14 May 2023

Dear Dr. Debes,

We are delighted to inform you that your manuscript, "Prioritizing interventions for cholera control in Kenya, 2015-2020," has been formally accepted for publication in PLOS Neglected Tropical Diseases.

Best regards,

Shaden Kamhawi

co-Editor-in-Chief

Paul Brindley

co-Editor-in-Chief
